# Occurrence, Trends, Management and Outcomes of Patients Hospitalized with Clinically Suspected Myocarditis—Ten-Year Perspectives from the MYO-PL Nationwide Database

**DOI:** 10.3390/jcm10204672

**Published:** 2021-10-12

**Authors:** Krzysztof Ozierański, Agata Tymińska, Marcin Kruk, Beata Koń, Aleksandra Skwarek, Grzegorz Opolski, Marcin Grabowski

**Affiliations:** 1First Department of Cardiology, Medical University of Warsaw, 02-097 Warsaw, Poland; krzysztof.ozieranski@gmail.com (K.O.); S073784@student.wum.edu.pl (A.S.); grzegorz.opolski@gmail.com (G.O.); grabowski.marcin@me.com (M.G.); 2National Health Fund, 02-528 Warsaw, Poland; Marcin.Kruk@nfz.gov.pl (M.K.); Beata.Kon@nfz.gov.pl (B.K.)

**Keywords:** cardiomyopathy, children, endomyocardial biopsy, epidemiology, heart failure, mortality

## Abstract

The epidemiology of myocarditis is unknown and based mainly on small single-centre studies. The study aimed to evaluate the current incidence, clinical characteristics, management and outcomes of patients hospitalized due to myocarditis in a general population. The study was registered in ClinicalTrials.gov (NCT04827706). The nationwide MYO-PL (the occurrence, trends, management and outcomes of patients with myocarditis in Poland) database (years 2009–2020) was created to identify hospitalization records with a primary diagnosis of myocarditis according to the International Classification of Diseases and Related Health Problems, 10th Revision (ICD 10), derived from the database of the national healthcare insurer. We identified 19,978 patients who were hospitalized with suspected myocarditis for the first time, of whom 74% were male. The standardized incidence rate of myocarditis ranged from 1.15 to 14 per 100,000 people depending on the age group and was the highest in patients aged 16–20 years. The overall incidence increased with time. The performance of the recommended diagnostic tests (in particular, endomyocardial biopsy) was low. Relative five-year survival ranged from 0.99 to 0.56—worse in younger females and older males. During a five-year follow-up, 6% of patients (3.7% and 6.9% in females and males, respectively) were re-hospitalized for myocarditis. Surprisingly, females more frequently required hospitalization due to heart failure/cardiomyopathy (10.5%) and atrial fibrillation (5%) than compared to males (7.3% and 2.2%, respectively) in the five-year follow up. In the last ten years, the incidence of suspected myocarditis increased, particularly in males. Survival rates for patients with myocarditis were worse than in the general population. Management of myocarditis requires significant improvement.

## 1. Introduction

Myocarditis is a major cause of heart failure and sudden cardiac death, mainly in children and young adults [1,2]. Myocarditis also constitutes a serious diagnostic and therapeutic problem due to the absence of knowledge gained via systematic investigation. Epidemiological data on the true incidence of myocarditis are lacking, and myocarditis is probably significantly underdiagnosed. The majority of information on the epidemiology, clinical characteristics and outcomes of myocarditis is derived from small single-centre studies [3]. What is more, the published studies offer conflicting results because of the wide variation in diagnostic criteria, hampering accurate estimations of the natural history of myocarditis.

It is important to design multicentre registries for assessing the myocarditis burden on a national or regional level, e.g., the Multicenter Lombardy Registry, which assessed the characteristics, in-hospital management and long-term outcomes of patients with acute myocarditis [4].

Large-scale databases are of particular importance as they may provide relevant information on population trends, demographics and outcomes for a given disease, including the tendencies occurring over a relatively long study period. National data are frequently published with regard to myocardial infarction or heart failure, but there are only single studies on myocarditis. The inpatient database in the United States presented an increasing incidence of hospitalizations due to myocarditis in the years 2007–2014, both in men and women [5]. The Global Burden of Disease 2016 Study (GBD2016) showed a 9% increase in all-age deaths due to myocarditis over a ten year period [6]. GBD2016 also highlighted that observed levels of years of life lost due to cardiomyopathy and myocarditis were much higher than expected, particularly in Eastern and Central Europe [6]. The recently published GBD 2019 showed that disability-adjusted life years and deaths due to cardiomyopathy and myocarditis have significantly increased over the past 30 years [7].

The presented data indicate a pressing need for actual populational data on myocarditis.

Therefore, a nationwide MYO-PL (the occurrence, trends, management and outcomes of patients with myocarditis in Poland) database combining information on all patients with myocarditis was created. In the current study, we aimed to evaluate the incidence, clinical characteristics and outcomes of patients with a hospital-based diagnosis of myocarditis in the last ten years in Poland.

## 2. Materials and Methods

In this retrospective-prospective study, we used data from the National Health Fund (NHF), which is the only public healthcare insurer in Poland. The NHF reimburses medication and healthcare services provided by healthcare providers (both public and private) with public funds collected from health insurance premiums. In Poland, public health insurance is obligatory for almost all Poles—in December 2019, 88.4% of approximately 38 million Poles had public health insurance and were entitled to obtain healthcare services and medication reimbursed by the NHF. Previously, more data were published based on the NHF database regarding the incidence of acute myocardial infarction [8].

Based on NHF claims data, we derived the healthcare services reported over the years 2009–2020 with a diagnosis of myocarditis—hospitalizations reported with codes I40, I40.0, I40.1, I40.8, I40.9, I41, I41.0, I41.1, I41.2, I41.8, I51.4 and B33.2 according to the International Classification of Diseases and Related Health Problems, 10th Revision (ICD-10). The diagnostic criteria of myocarditis (based on international ICD-10 codes) were clinician-dependent, reflecting routine clinical practice. No other specific inclusion-exclusion criteria were applied, as this is a population database. It should be noted that the criteria for the diagnosis of myocarditis (as well as access to advanced diagnostic procedures, i.e., cardiac magnetic resonance (CMR) and endomyocardial biopsy) have changed over time, making the diagnosis of myocarditis usually a diagnosis of exclusion. Myocarditis should be confirmed by endomyocardial biopsy; however, this is performed only in selected centres worldwide and still not in all patients.

We narrowed the dataset to newly diagnosed myocarditis (first hospitalization), i.e., we included patients for whom no information about the diagnosis of myocarditis was reported in the 400 days preceding hospitalization. For such a group of patients and to establish the baseline characteristics of the patients, data were analyzed 400 days back from the initial diagnosis of myocarditis. Moreover, in-hospital and long-term outcomes were analyzed, including all-cause mortality as well as the occurrence of selected diseases (defined as receiving a service where the selected ICD-10 code was reported) and selected procedures, defined with codes according to International Classification of Diseases, 9th Revision, Clinical Modification (ICD-9-CM) (all ICD codes used in this analysis are presented in a Appendix A). For the follow up, at least a six month period was required. Thus, only patients with a diagnosis of myocarditis between January 2011 and December 2019 were included in the analysis.

To show differences related to the age of patients hospitalized due to myocarditis, the baseline characteristics and long-term outcomes were assessed with regard to age groups.

No ethics approval was required for this study, as it involved the analysis of administrative data. The study complies with the Declaration of Helsinki.

In the study, data from the Central Statistical Office of Poland were also used to refer the obtained results to the population of Poland and to obtain life tables for survival analysis.

### Statistical Analysis

The results were presented as means (and standard deviations) or medians (and quartiles) for continuous variables. Ordinal variables were presented as percentages. Associations between study parameters were analyzed using a Pearson chi-square test and a t-test. The observed survival rate was analyzed using the Kaplan–Meier estimates. The relative survival rate (with 95% CIs) was calculated using the Hakulinen method employing single age-specific, year-specific, and sex-specific life tables for the general Polish population. A *p*-value less than 0.05 was considered significant. All tests were two-tailed. Statistical analysis was carried out using R software, version 3.6.1.

## 3. Results

### 3.1. Study Population and Clinical Characteristics

During the study period (2011–2019), there were 19,978 patients hospitalized due to myocarditis. The median age of the total cohort was 33 years (32 and 46 years in males and females, respectively). The majority of the patients were male (74%, *n* = 14 870), regardless of the age group (75.4% and 74.2% in patients aged ≤20 and >20 years (*p* = 0.14), respectively).

The incidence of myocarditis was the highest in patients aged 16–20 years (up to 14/100000 in 2016) (Table 1). Two peaks in the overall incidence of myocarditis could be distinguished—in children aged 0–5 years and then in young adults starting from 16–20 years and with a slow decrease from those aged 31–40 years (Figure 1). The overall incidence increased over time, but was driven by a substantial increase in myocarditis incidence in males (Figure 2). The proportion of males was higher in all age groups except patients aged 71–80 and 81+ (Figure 1). In contrast, in females, a slight decrease in the incidence rate was observed over time. Interestingly, the sex distribution of patients with myocarditis was, nevertheless, age-dependent (Figure 1). The proportion of females to males tended to increase with age and was the highest in the oldest age groups.

Most of the clinical characteristics differed between patients aged ≤20 and >20 years. Chronic diseases, except for a history of asthma, were more common in the older group. Diagnoses of otolaryngologic and ophthalmic or digestive infectious disease in the prior 6 months were observed more frequently in the younger group. Patients aged ≤20 years were more likely to suffer from bradycardia or tachycardia/palpitations, while atrial fibrillation and ventricular tachycardia were more common in those aged >20 years. Most patients were hospitalized in general/cardiology or other hospital wards. On the other hand, approximately 2% of patients required hospitalization in an intensive (cardiac) care unit—more likely for patients ≤20 years than for the older group. Reported use of diagnostic procedures, particularly CMR and endomyocardial biopsy, was very low. The main clinical characteristics and management of patients with myocarditis are presented in Table 2.

### 3.2. Outcomes

A total of 494 (2.5%) patients died during the index hospitalization (28 (0.8%) and 466 (2.9%) in the groups aged ≤20 and >20 years, respectively). The observed five year survival rate ranged from 0.98 in males aged 0–20 years to 0.26 in males older than 80 years. The relative five-year survival rate, however, ranged from 0.987 to 0.56 in these age-sex groups. Observed and relative survival rates in relation to sex and age are shown in Table 3 (five-year follow up) and Figure 3 (ten-year follow up). Relative survival was associated with age and sex and was higher in younger male age groups than in younger female age groups, while the rate was better in females than in males in older patients. A decline with age and follow-up time in both observed and relative survival rates was observed. Relative survival was substantially less affected by the patient’s age than observed survival.

During a five-year follow-up, 6% of the study group (3.7% and 6.9% in females and males, respectively) were readmitted to hospital due to myocarditis. Interestingly, females required hospitalization due to heart failure/cardiomyopathy (10.5%) and atrial fibrillation more frequently (5%) than men (7.3 and 2.2, respectively) in a five-year follow up.

## 4. Discussion

Myocarditis is an inflammatory heart disease caused by multiple infectious, non-infectious factors and immune-mediated factors [1]. It presents a challenge in modern cardiology because of the deficit of systematic knowledge of diagnostic and therapeutic modalities. Diagnosis of myocarditis is very demanding and can be reported with multiple ICD-10 codes (I40, I40.0, I40.1, I40.8, I40.9, I41, I41.0, I41.1, I41.2, I41.8, I51.4 and B33.2). The published studies offer conflicting results because of the wide variation in diagnostic criteria (i.e., biopsy proven or not; type of cellular infiltrations; underlying etiology; virus-positive or virus-negative myocarditis; active or chronic inflammation). This was particularly emphasized during the severe acute respiratory syndrome coronavirus 2 pandemic, when many papers with different definitions of myocarditis were published, showing a distorted picture of the true scale of the disease [9,10]. Moreover, in recent years, no large-scale analyses have been conducted, meaning that the current epidemiology of myocarditis is unknown.

This nationwide study includes the unique data of all patients hospitalized with clinician-dependent diagnosis of myocarditis in Poland in the last ten years. It provides new and relevant information with regard to population trends, demographics and the outcomes of patients with myocarditis met in routine clinical practice. This study is particularly important as most of the demographic data for myocarditis derives from small cohort studies hampering accurate estimations of the natural history of myocarditis. During the ten year observation period, myocarditis occurred in patients of all ages with clear predominance in patients aged 16–30 years (Table 1). The overall incidence ranged from 1.15 to 14.07 per 100,000 residents with two peaks—in children aged 0–5 years and then in young adults aged 16–30 years (Figure 1). The age distribution of female patients with myocarditis was notably more stable when compared to male patients. Importantly, we observed an overall increase in myocarditis incidence over time, mainly driven by a substantial increase in myocarditis incidence in male patients (Figure 1).

Previously published data presented similar results that myocarditis was significantly more common in male (approximately 73%) than in female patients [11]. What is more, male patients with myocarditis were noticeably younger than female patients (mean age approximately 34 vs. 49 years, respectively) [11]. Our study showed that the vast majority of patients with myocarditis were male for both children and adults (approximately 74% in both groups aged ≤20 and >20 years). What is more, at the moment of diagnosis, males were several years younger than females (32 vs. 46 years, respectively). Previous studies demonstrated that the incidence of myocarditis was even higher than in our database (1.8–18 per 100,000) [12,13]. There are still no reliable explanations for the increasing incidence of myocarditis. Perhaps it is associated with greater accessibility, mainly to non-invasive tests, such as echocardiography and CMR, or greater awareness of myocarditis. On the other hand, the increased incidence might be related to the greater number of infectious, non-infectious and immune-mediated factors causing myocarditis. To address this question, etiological and pathological analyses should be performed in adequately designed clinical trials.

According to the current criteria of the European Society of Cardiology (ESC), myocarditis should be suspected based on the clinical picture and additional tests (ECG, echocardiography, CMR, troponins) [1]. The diagnosis should be confirmed by endomyocardial biopsy (diagnostic gold standard), but this procedure is infrequently used. Our study showed that the application of ESC criteria for the diagnosis of myocarditis in clinical practice is rare. Non-invasive tests were performed infrequently (troponins—41.3%; echocardiography—81.1%; CMR—16.4%). Ischemic etiology was verified only in 30.9% of patients—significantly more frequently in patients aged >20 years—which was understandable. Endomyocardial biopsy was reported only for a marginal proportion of patients (0.3% vs. 0.8% in patients aged ≤20 and >20 years, respectively). The Nationwide US Inpatient Database (1998–2013) identified 22,299 hospitalization records with a diagnosis of myocarditis and of those, only 798 patients (3.6%) underwent endomyocardial biopsy [12]. What is more, the use of this procedure has been reported as significantly decreasing over time, despite a significant upward trend in the total number of patients with myocarditis [12]. The current ESC recommendations were published in 2013; however, the performance of the recommended diagnostic tests in the observed time period was very low. In particular, the use of endomyocardial biopsy seems to be insufficient. Nowadays, endomyocardial biopsy with current histologic, immunohistochemical and molecular methods provides a diagnosis of certainty in clinically suspected myocarditis and improves differential diagnosis in idiopathic cardiomyopathy or cardiac arrhythmias. It also allows clinicians to establish adequate treatment and monitor the therapy. This indicates that the diagnosis of myocarditis with the use of non-invasive and invasive procedures requires significant improvement.

Myocarditis has been shown in post-mortem studies to be a major cause (up to 42% of cases) of sudden and unexpected death in children and young adults [14,15]. In contrast, a recently published study on autopsies reported that 6% of 14,294 sudden deaths were assigned as being caused by myocarditis [16]. These differences are likely explained by the heterogenicity of the study populations and differences in sudden death, as well as myocarditis definitions and classifications.

Comparison of our database with the Multicenter Lombardy Registry of patients (*n* = 684) with acute myocarditis, diagnosed either by endomyocardial biopsy or increased troponin plus oedema and late gadolinium enhancement in CMR, shows similar results in the mortality during index hospitalization—2.5 and 3.2%, respectively [4]. Despite the fact that the cited study included patients with fulminant myocarditis (associated with worse prognosis), the difference in hospital mortality is small. This difference may be the result of different inclusion criteria (e.g., exclusion of patients older than 70 years and older than 50 years of age without coronary angiography) and diagnostic procedures, on the basis of which the diagnosis of acute myocarditis was made. Interestingly, a different study on clinically diagnosed acute myocarditis performed on a smaller cohort (*n* = 322), reported no deaths [17]. This may be due to the relatively high left ventricular ejection fraction (LVEF) in the study cohort. The mean LVEF at presentation was 54 ± 9%. Another prospective study conducted on patients with clinically diagnosed myocarditis (*n* = 187) investigated mortality in the group of fulminant myocarditis (18.2%) vs. non-fulminant myocarditis patients (0%) [18].

In patients with biopsy-proven myocarditis in long-term observation (the median follow up of 4.7 years), all-cause mortality was 19.2%, while sudden death occurred in 9.9% of cases [19]. Worse outcomes were reported in patients with symptomatic heart failure, reduced LVEF, presence of late contrast enhancement in CMR, malignant ventricular arrhythmias and/or confirmed viral infection in endomyocardial biopsy [20]. In our study, patients with myocarditis had worse survival rates in all age groups than their counterparts in the general population. Our study included a broad spectrum of clinical presentations of myocarditis; therefore, the presented survival rate comprised an average of patients with mild to severe clinical status. The older the age group, the worse the prognosis in comparison to the sex-age-matched general population, although the relative five year survival rate differed in terms of age and sex: In younger age groups, it was better in male patients, while in older age groups female patients had higher survival rates than male patients. It was previously shown that the male sex was associated with a worse course of myocarditis [21]. In the post-mortem study, it was also reported that male patients had a significantly higher risk of sudden death related to myocarditis when compared to female patients [16]. On the other hand, there exists a study showing no sex difference in the rate of major adverse cardiovascular events (MACE) defined as composite of occurrence of congestive heart failure, non-sustained ventricular tachycardia and/or sustained ventricular tachycardia [22]. The vast majority of MACE patients reported in this study had either non-sustained or sustained ventricular tachycardia, and there were no deaths. The exact reason for the mentioned sex differences in the incidence of myocarditis and associated outcomes is unknown.

The survival of patients with myocarditis might be influenced by disease-specific treatment (there is available data about certain effective forms of immunosuppression for biopsy-proven myocarditis) [1,23,24]. Spontaneous or treatment-induced improvement of left ventricular function was observed within a few months of the disease onset in 40–50% to 90% of patients, respectively [3,25]. However, no therapy can yet be approved because of the lack of adequately conducted clinical trials. Data obtained from nationwide databases clearly demonstrate that myocarditis constitutes an emerging challenge for healthcare systems and decisive steps need to be taken to stimulate research in the field.

### Limitations

The inclusion of real-life patients allowing for population analyses is an important advantage of this nationwide database; however, it has several limitations that have to be acknowledged. First, in terms of baseline clinical characteristics, the data were limited to 400 days prior to inclusion in the study; therefore, it might be incomplete. Second, errors in ICD-9 and ICD-10 coding and documentation misclassification bias cannot be excluded. Moreover, it is possible that not all diagnoses and procedures may have been reported. There was also an absence of detailed clinical and laboratory data because of the nature of the database. Third, as mentioned above, true myocarditis should be confirmed based on endomyocardial biopsy. However, in the vast majority of cases, it was clinically suspected myocarditis with infrequent use of CMR.

## Figures and Tables

**Figure 1 jcm-10-04672-f001:**
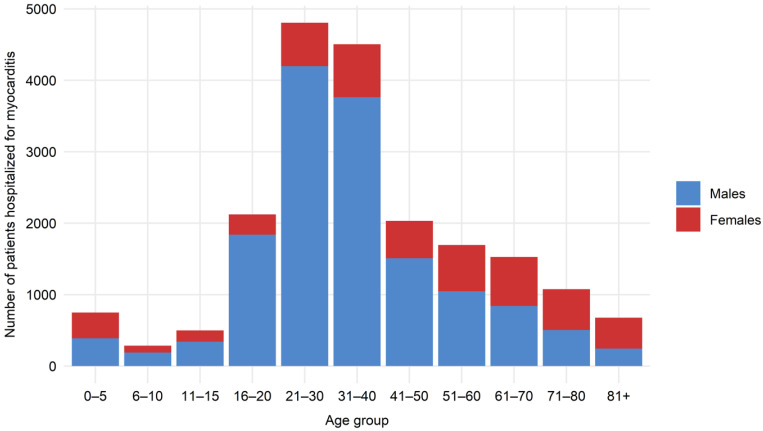
Age and gender distribution of all patients hospitalized for myocarditis in Poland in years 2011–2019. Red—females; blue—males.

**Figure 2 jcm-10-04672-f002:**
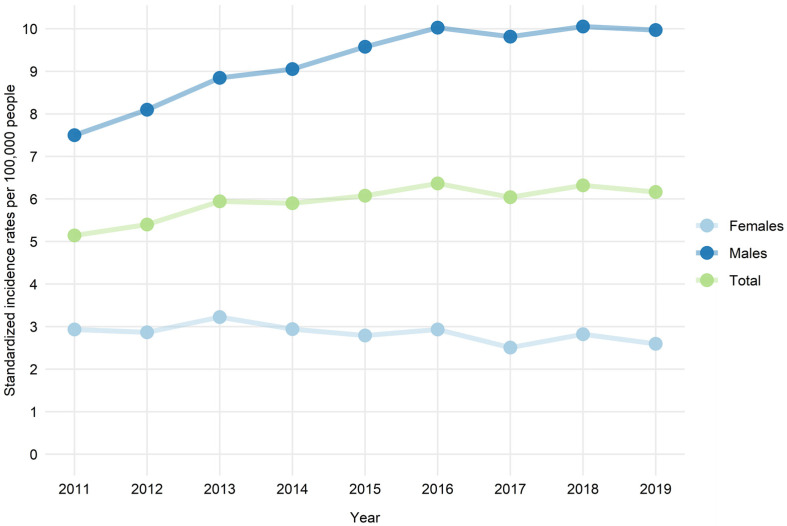
Age-standardized hospitalization rates for myocarditis of males and females by the number of residents in Poland in the years 2011–2019.

**Figure 3 jcm-10-04672-f003:**
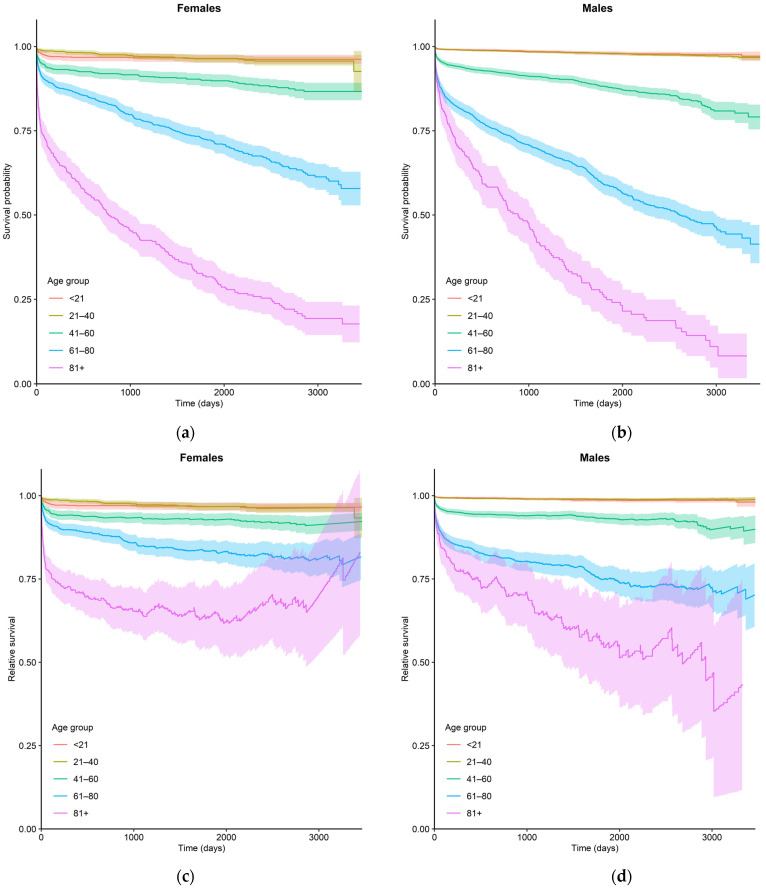
Observed (**a**,**b**) and relative (**c**,**d**) ten-year survival rates of patients hospitalized for myocarditis in relation to sex and age.

**Table 1 jcm-10-04672-t001:** Incidence of patients with myocarditis by the number of 100,000 residents in Poland in a given age group in the years 2011–2019.

Age Group (Years)	Year									*p* Value ^a^
	2011	2012	2013	2014	2015	2016	2017	2018	2019	
0–5	4.27	4.06	4.53	4.14	3.30	2.99	3.24	3.06	2.13	0.000
6–10	1.50	0.99	2.09	1.59	1.09	2.13	1.15	2.42	1.54	0.002
11–15	1.81	1.87	3.37	3.44	3.05	3.45	3.59	3.44	2.88	0.002
16–20	8.79	8.95	11.55	10.60	11.39	14.07	12.51	12.77	12.83	0.000
21–30	7.05	8.72	8.52	10.35	10.41	11.18	10.91	11.09	11.08	0.000
31–40	6.71	7.22	7.35	8.08	8.79	8.39	8.72	8.42	8.99	0.000
41–50	2.94	3.39	4.61	3.89	4.82	4.83	4.82	5.71	5.12	0.000
51–60	3.55	3.41	3.66	3.50	3.00	3.61	3.56	3.91	3.73	0.506
61–70	4.85	3.93	4.51	3.39	3.66	3.91	3.36	3.51	3.29	0.001
71–80	6.85	6.42	6.32	5.10	5.72	4.50	2.87	3.78	3.90	0.000
81+	7.14	7.63	7.66	6.15	6.07	5.11	3.60	3.75	3.61	0.000

**^a^** A Chi-square test for the independence of the number of patients with myocarditis in a given age group by year.

**Table 2 jcm-10-04672-t002:** Clinical characteristics and performed diagnostic procedures in hospitalized patients with myocarditis.

Variable	Total (*n* = 19,978)	Age ≤20 Years (*n* = 3659)	Age >20 Years (*n* = 16,319)	*p*-Value *
Demographics				
Males, *n* (%)	14870 (74.4)	2759 (75.4)	12111 (74.2)	0.14
Median Age (IQR) - Total	33 (23–50)	17 (8–19)	37 (29–56)	-
- Females	46 (27–66)	10 (1–16)	54 (36–70)	-
- Males	32 (23–43)	17 (13–19)	35 (28–47)	-
Management				
Hospital ward, *n* (%) - Cardiology unit	11305 (54.4)	1492 (38.4)	9813 (58.1)	<0.0001
- General ward	5087 (24.5)	1817 (46.8)	3270 (19.4)	<0.0001
- Intensive care unit	263 (1.3)	64 (1.6)	199 (1.2)	0.02
- Intensive cardiac care unit	152 (0.7)	32 (0.8)	120 (0.7)	0.52
- Other	3974 (19.1)	479 (12.3)	3495 (20.7)	<0.0001
Diagnostic procedures, *n* (%) - C-reactive protein **	8332 (41.7)	1494 (40.8)	6838 (41.9)	0.24
- Brain natriuretic peptides **	2754 (13.8)	475 (13.0)	2279 (14.0)	0.13
- Troponins **	8254 (41.3)	1349 (36.9)	6905 (42.3)	<0.0001
- Echocardiography ***	16206 (81.1)	3188 (87.1)	13018 (79.8)	<0.0001
- Cardiac Magnetic Resonance ***	3284 (16.4)	563 (15.4)	2721 (16.7)	0.06
- Endomyocardial biopsy ***	142 (0.7)	12 (0.3)	130 (0.8)	0.003
- Endomyocardial biopsy or heart catheterization ***	251 (1.3)	40 (1.1)	211 (1.3)	0.37
- Coronary angiography (invasive or computed tomography) ***	6172 (30.9)	270 (7.4)	5902 (36.2)	<0.0001
Medical history (within up to last 400 days)				
Cardiac Arrhythmias, *n* (%) - Atrial Fibrillation	788 (3.9)	7 (0.2)	781 (4.8)	<0.0001
- Tachycardia, palpitations	266 (1.3)	63 (1.7)	203 (1.2)	0.03
- Bradycardia	22 (0.1)	12 (0.3)	10 (0.1)	<0.0001
- Atrial extra beat	20 (0.1)	6 (0.2)	14 (0.1)	0.29
- Ventricular extra beat	53 (0.3)	14 (0.4)	39 (0.2)	0.18
- Paroxysmal tachycardia	206 (1.0)	19 (0.5)	187 (1.1)	0.001
- Ventricular tachycardia	48 (0.2)	2 (0.1)	46 (0.3)	0.02
- Ventricular fibrillation	18 (0.1)	2 (0.1)	16 (0.1)	0.63
Chronic coronary syndrome, *n* (%)	1105 (5.5)	3 (0.1)	1102 (6.8)	<0.0001
Heart failure, *n* (%)	1173 (5.9)	19 (0.5)	1154 (7.1)	<0.0001
Hypertension, *n* (%)	3440 (17.2)	53 (1.4)	3387 (20.8)	<0.0001
Diabetes, *n* (%)	817 (4.1)	17 (0.5)	800 (4.9)	<0.0001
Stroke or transient ischemic attack, *n* (%)	279 (1.4)	1 (0)	278 (1.7)	<0.0001
Chronic kidney disease, *n* (%)	210 (1.1)	2 (0.1)	208 (1.3)	<0.0001
Asthma, *n* (%)	915 (4.6)	227 (6.2)	688 (4.2)	<0.0001
Autoimmune disease, *n* (%)	267 (1.3)	28 (0.8)	239 (1.5)	0.001
Psychiatric diseases, *n* (%)	438 (2.2)	44 (1.2)	394 (2.4)	<0.0001
Infectious disease within last 6 months, *n* (%) Otolaryngologic and eye	6707 (33.6)	1548 (42.3)	5159 (31.6)	<0.0001
Central nervous system	16 (0.1)	4 (0.1)	12 (0.1)	0.71
Respiratory	2867 (14.4)	503 (13.7)	2364 (14.5)	0.26
Digestive	947 (4.7)	200 (5.5)	747 (4.6)	0.02
Urogenital	149 (0.7)	30 (0.8)	119 (0.7)	0.64
Sepsis	73 (0.4)	14 (0.4)	59 (0.4)	0.97
Other	1152 (5.8)	211 (5.8)	941 (5.8)	>0.999

* Comparison between patients aged ≤20 and >20 years old. ** within the index hospitalization. *** within last or proceeding 6 months from diagnosis of myocarditis. IQR—interquartile range; *n*—number.

**Table 3 jcm-10-04672-t003:** Observed and relative survival by sex and age.

Gender	Survival	Age Group				
		0–20	21–40	41–60	61–80	81+
Observed survival						
Males	1 year	0.990 (0.987–0.994)	0.989 (0.986–0.991)	0.934 (0.924–0.944)	0.802 (0.781–0.823)	0.663 (0.604–0.722)
	3 year	0.985 (0.980–0.990)	0.984 (0.981–0.987)	0.911 (0.899–0.922)	0.697 (0.671–0.722)	0.419 (0.353–0.485)
	5 year	0.980 (0.974–0.986)	0.979 (0.976–0.983)	0.881 (0.867–0.896)	0.589 (0.559–0.619)	0.256 (0.192–0.320)
Females	1 year	0.969 (0.958–0.980)	0.983 (0.976–0.990)	0.930 (0.915–0.944)	0.869 (0.850–0.888)	0.620 (0.575–0.666)
	3 year	0.966 (0.954–0.978)	0.970 (0.960–0.980)	0.913 (0.896–0.929)	0.784 (0.761–0.808)	0.430 (0.382–0.478)
	5 year	0.965 (0.952–0.977)	0.964 (0.952–0.975)	0.900 (0.882–0.918)	0.722 (0.695–0.750)	0.316 (0.268–0.364)
Relative survival						
Males	1 year	0.994 (0.991–0.997)	0.992 (0.989–0.994)	0.946 (0.937–0.956)	0.845 (0.823–0.866)	0.764 (0.696–0.832)
	3 year	0.990 (0.985–0.994)	0.99 (0.987–0.993)	0.941 (0.929–0.953)	0.800 (0.771–0.829)	0.648 (0.546–0.749)
	5 year	0.987 (0.981–0.992)	0.989 (0.986–0.993)	0.933 (0.918–0.948)	0.750 (0.712–0.787)	0.560 (0.421–0.698)
Females	1 year	0.971 (0.960–0.982)	0.984 (0.977–0.991)	0.941 (0.927–0.955)	0.896 (0.878–0.915)	0.717 (0.665–0.768)
	3 year	0.969 (0.957–0.980)	0.972 (0.963–0.982)	0.931 (0.915–0.947)	0.850 (0.824–0.875)	0.643 (0.572–0.714)
	5 year	0.967 (0.955–0.979)	0.967 (0.956–0.978)	0.928 (0.910–0.946)	0.833 (0.802–0.864)	0.637 (0.542–0.733)

## Data Availability

Data available on request.

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
