# Peer review of "Occurrence, Trends, Management and Outcomes of Patients Hospitalized with Clinically Suspected Myocarditis—Ten-Year Perspectives from the MYO-PL Nationwide Database"

_jcm, 2021, doi:10.3390/jcm10204672_

Round 1

Reviewer 1 Report

my comments  are as follows,  please review it carefully.

Review:

Occurrence, Trends, management and outcomes of patients hospitalized with clinically suspected myocarditis-ten year- per- spectives from the MYO-PL nationwide database.

Krzysztof et al presented a unique data of patients with acute myocarditis hospitalized between 2009-2020 based on the ICD-10. They identified  19978 patients with a first time suspected myocarditis, of whom 74% where males, myocarditis had a higher incidence among young age groups

In this analysis they concluded that the performance of the recommended diagnostic tests was low. In addition relative 5 year survival varied significantly among the different age and gender groups , thus  young females as well as older males had the worst survival rates.

The authors found that at 5 years of follow-up the rehospitalization rate due to myocarditis was 6% and was higher among males. Importantly females were more likely to have been hospitalized due to heart failure/cardiomyopathy and AFIB.

Importantly the authors showed that during the last ten years there was an increased incidence of myocarditis , in addition  survival rates among myocarditis patients were worse than the general population.

Comments for the authors:

  • Introduction is well presented and very well written , but there should be relation to some of the important studies in the field for example one large multicenter study of patients with clinically suspected myocarditis study presented by Ammirati et al ( clinical presentation and outcome in a contemporary cohort of patients with acute myocarditis-Circulation 2018). This is my only comment regarding this section.

2- Materials and methods: very well presented and detailed no comments  on this section.

3- Results section: very well study and important results , one important aspect is not focusing on the adults, but on all age groups even children between 0-5 which is significant point in comparison with all contemporary studies in the field. In addition they demonstrated a unique distribution of myocarditis across age groups also importantly the authors showed a significant change in the proportion of males compared to females with aging.

Unique important point regarding in the results is the age as well as sex  based differences in the clinical presentation of patients.

Minor comment: In this study there is a relatively not small  percentage of death during the index hospitalization, please explain and compare this to other studies in the field , as it is important to compare it with the Lumbardy registry of patients with clinically suspected myocarditis , Ammirati et al ( clinical presentation and outcome in a contemporary cohort of patients with acute myocarditis-Circulation 2018). In addition our group published  a review of patients with clinically suspected myocarditis  with no deaths during the indexed hospitalization (Younis et al American journal of Medicine).  However also other studies in the field showed a significant rate of death or heart transplant of 25% among patients with fulminant myocarditis ( Survival and left ventricular function changes in fulminant versus non fulminant acute myocarditis. Circulation 2018).

Minor comment 2:  the authors concluded that the survival rate was higher among males in younger age groups , whereas in older age groups survival rates were better among females, please discuss this as for exam[le our group published a study in the American journal of cardiology (Sex-based differences in characteristics and in-hospital outcomes among patients with diagnosed acute myocarditis 2020), in this study we did not find significant differences in in hospital outcomes of patients with myocarditis.

Overall very important and valuable study please just compare your results with the above mentioned studies.

Reviewer 2 Report

The study is an interesting population study with the inherent limitations of the study's design.

The presentation is high quality and the statistics are well designed. The definition of myocarditis is highly hetereogeneous representing real world clinical practice.

Just one comment: although the authors report that multivariable analysis was performed, no such results are presented. 
